# Facilitating the natural semi-drying of oily sludge by changing the form of water

**Yucheng Liu** [1,2]*, **Maoren Wang**[1], **Mingyan Chen**[1,2], **Meng Zhu**[1], **Maoqi Liao**[1]

**1** College of Chemistry and Chemical Engineering, Southwest Petroleum University, Chengdu, Sichuan, P R of China, **2** Research Institute of Industrial Hazardous Waste Disposal and Resource Utilization, Southwest Petroleum University, Chengdu, Sichuan, P R of China

* rehuo2013@sina.cn

**Data Availability Statement:** All relevant data are within the manuscript and its Supporting Information files.

**Funding:** This work was financially supported by Sichuan Science and Technology Plan Key

## Abstract

Reducing the water content of oily sludge is essential for the disposal of it. Despite conditioning and solid-liquid separation, the water content of oily sludge generally exceeds 65%. A large amount of this water exists in the form of emulsified and bound water, reducing the efficiency of water removal during the natural semi-drying process of oily sludge. To shorten the time required for natural semi-drying, this study applied an orthogonal test to screen an oily sludge modified material (OSM). The effect and mechanism of OSM on the natural semi-drying of oily sludge were studied using a thermal gravimetric analyzer (TGA), scanning electron microscope (SEM), surface tension measurement, and microscopic observations. The results show that when the ambient temperature exceeded 10°C, the OSM increased in mass by 8–10%, and the time required for the natural semi-drying of oily sludge was shortened from 15 days to less than 5 days. OSM can rupture the emulsion, reduce the surface tension, convert the emulsion and bound water in the oily sludge into free water, and accelerate the rate of water migration, diffusion, and natural evaporation from the inside of the oily sludge to the surface and air. The research results show that changing the form of water can speed up the drying of oily sludge, creating positive economic benefits in the process of oily sludge reduction and recycling.

## Introduction

A large amount of oily sludge is produced during crude oil exploitation, transportation, storage, and crude oil refining [1]. Records show that China's refineries generate an average of more than $1.0 \times 10^6$ tons of oily sludge per year. Many countries have identified oily sludge as a Dangerous Waste because of its many hazardous ingredients [2]. Several treatment methods have been widely studied and applied, including chemical extraction [3], thermal desorption [4], microwaving [5], incineration [6–8], and microbial degradation [9, 10]. However, the oily sludge that remains after conditioning and solid-liquid separation generally has high moisture content and low oil content (e.g., crude oil 1.8 wt%, water 65.8 wt%). The water mainly exists in the form of bound water, emulsified water, and a small amount of free water. Bound water refers to water that is attracted to the internal pore surfaces and external surfaces of oily sludge

Research and Development Project Funding
(2018GZ0421).

**Competing interests:** The authors have declared
that no competing interests exist.

particles by charged molecules. Emulsified water refers to oil-water emulsion, which is a water-in-oil, oil-in-water, or a dispersion system with multiple types of emulsification. Free water refers to water that can flow freely.

The preparation of sintered brick and non-fired bricks is considered a safe and resource-aware method of disposing of oily sludge and has been applied in Sichuan, Chongqing, and other Chinese provinces [11]. Due to the limitations in plasticity, the shrinkage rate, and drying sensitivity coefficient, the oily sludge used to produce bricks requires drying to reduce the moisture content to below 50%.

Drying methods include paddle drying [12], microwave drying [13], solar drying [14], and natural drying. Paddle drying uses hot water, steam, or heat-conducting oil as a heat source for indirect heat exchange. Microwave drying has been widely used in the physical processing of food, paper, wood, and products in other industries. In addition to safety concerns, the high cost of these approaches consumes significant energy. Solar dryers may be the simplest and least expensive technology to reduce sludge weight [15]. These dryers mainly include greenhouse and fluidized bed modes. However, the kinetics of solar drying fluctuate, as the supply of solar energy varies [16]. Compared with paddle drying and microwave drying, natural air drying, similar to solar drying, is the most economical and effective method for reducing the moisture content of oily sludge to a semi-dry state.

Drying methods can be divided into two types based on the moisture content: total drying and semi-drying. Total drying decreases the oily sludge moisture content to 10–15% or less. Semi-drying decreases the moisture content to 40–50%. Natural semi-drying refers to placing the oily sludge at rest at room temperature. Most of the water in the oily sludge is then removed through natural diffusion and evaporation. This reduces the water content in the oily sludge to a new level of 40–50%.

Natural drying depends on many factors, including temperature, wind, air humidity, and oily sludge stacking thickness. It takes 15 days or more for oily sludge to naturally dry to a semi-dried state after flocculation and solid-liquid separation. Three factors are particularly important: (1) operating many storage sites increases overall enterprise costs. (2) the longer the storage time and the greater the amount of storage required is, the greater the risk is to safety and the environment. (3) gaseous emissions produce odour during the production process. These factors highlight the need to shorten the drying period.

To address these problems, it is important to explore and develop modified materials to accelerate the natural drying of oily sludge. For example, chemical agents can reduce the time of drying and the putrid smell of oily sludge [17]. Modified materials generally require the following characteristics: (1) have an inexpensive and widely available source; (2) be able to absorb the moisture from oily sludge and not react at high temperatures (i.e., it cannot be physically adsorbed or react with chemical agents to form crystalline water); (3) the reaction process must release heat and accelerate the natural evaporation of water; and (4) increase mass as little as possible to avoid increasing the total amount of oily sludge requiring transport.

A few studies have accelerated the drying of oily sludge during natural drying [18, 19]. However, studies have not examined the effect and mechanism of chemicals on natural semi-drying, especially with respect to the effect of the form of water on the drying rate. In this study, we examine how the form of water in an oily sludge can be modified using a chemical agent as a modified material (OSM) to speed up the drying process. This process facilitated the semi-dry state of oily sludge in 5 days. The semi-drying effect of OSM in oily sludge was evaluated using an orthogonal experiment, and the influencing factors of the semi-drying process were explored. Finally, the study analysed the mechanism involved in the drying process.

## Materials and methods

### Materials

The powder emulsifier in the OSM used for this study included sodium alpha-olefin sulfonate, aminosulfonic acid, and sodium dodecyl benzene sulfonate. The oxidant included $FeCl_3$ and $Fe_2(SO4)_3$. The inorganic cementitious material included CaO (effective content >98%), and Portland cement (industrial level). Other reagents included $CaCl_2$, $MgSO_4$, $NaSiO_4$, and $MgCl_2$. Of the reagents above, the Portland cement came from the Pengzhou Cement Factory in Sichuan Province in China. The remaining reagents came from the Chengdu Kelong Chemical Testing Factory in China.

This study was approved by Engineering Technology Research Institute of Xinjiang Oilfield Company. The oily sludge was collected from the Xinjiang oil field in China. The oily sludge was black, in the form of a semi-solid cake at ambient temperature and exuded a strong odour of light petroleum volatilization (**Fig 1**). The oil content of the sample was 2–6%, and the moisture content was 68–72%. The oil contained additives, such as polymers, that remained during flocculation.

### Experiment

**The principles of modified material screening.**    The main role of the OSM was to: (1) react with water in oil sludge; (2) convert the emulsified water into free water; (3) convert the bound water into free water; (4) generate an exothermic reaction, accelerating water evaporation; (5) enable a reaction to allow the oxidation to decrease the organic matter and reduce the ability to absorb water; and (6) apply other reagents to speed up the reaction time of the inorganic cementitious materials or increase water absorption. Based on the above design, the composition of the modified material included a powder emulsifier, oxidant, inorganic cementitious material, and other reagents. For each of the above 4 additive types of materials (agents),

# Figures

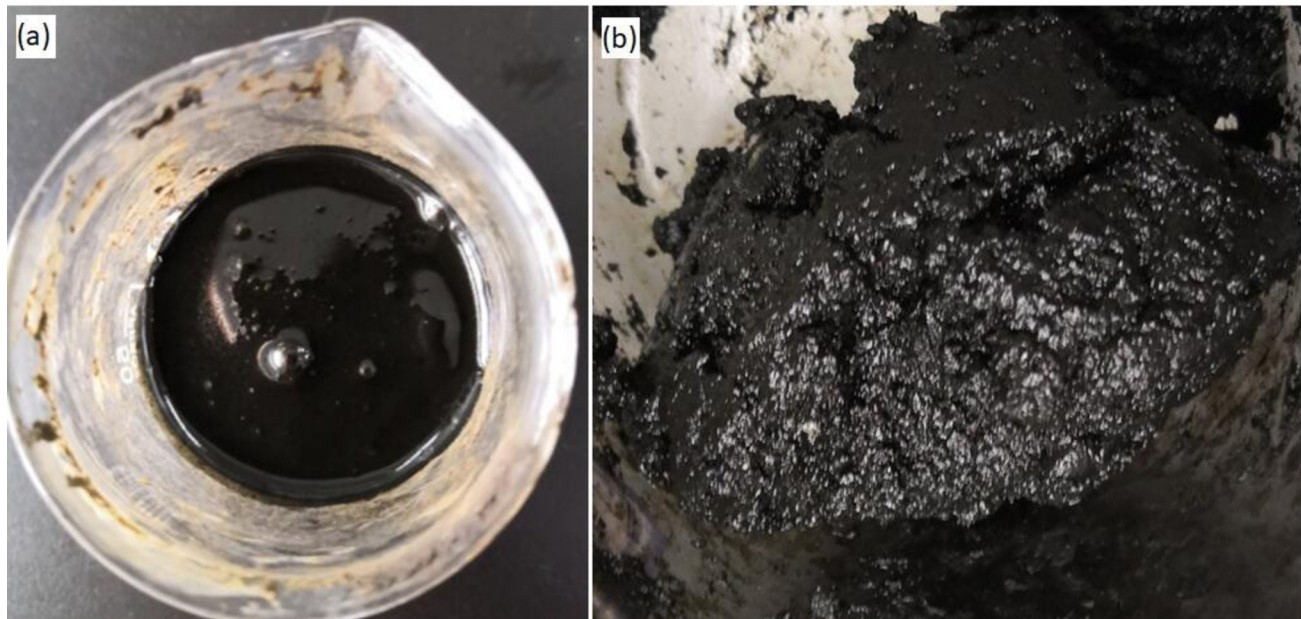

**Fig 1.** Oily sludge samples: crude oily sludge (a), after solid-liquid separation of oily sludge (b).

a single agent was tested at a time and was compared with the blank sample. The materials that resulted in a water content level higher than the blank sample were removed from further consideration. The additive material types resulting in the lowest moisture content was used for the tested OSM mixture used in the orthogonal experiments. Each data result presented in this article reflects the average of three parallel experiments. The standard error of the mean is reported to reflect the degree of dispersion.

**Initial screening.**   Each reagent was added to ($m_x$), g (FA200 electronic balance, sensing 0.0001 g, Shanghai Tianmei China) of oily sludge at a level of 10% of the mass. The sample was then stirred evenly and maintained at room temperature. The moisture content was measured after 5 days. The material with the lowest moisture content was selected as the raw material for the preliminary screening of the modified materials.

**Surface tension measurement.**   The oily sludge leachate was collected and added to the OSM. Surface tension (XZD-SP Beijing Harko, China) of the OSM was measured at room temperature with distilled water as a reference material.

**Moisture content measurement.**   (1) The sample moisture content was assessed by weighing an oil sludge sample ($m_0$) and then heating to a constant weight ($m_{0'}$) (constant temperature drying box 101-1AB Tianjin Taisite China) at 80˚C. The formula to calculate the moisture content is (noted as w%):

$$w\% = \frac{m_0 - m_{0'}}{m_0} * 100\% \tag{1}$$

(2) To assess the moisture content of the oily sludge after adding the chemical reagent, we assessed the mass of the beaker, added oily sludge, and added OSM or chemical reagent based on the mass percentage. The sample was naturally semi-dried at room temperature for a set period $t_1$, and the mass was then measured. After continuing to naturally semi-dry the sample for a set period $t_2$ at room temperature, the mass was measured again. The sample was then dried at 80˚C until it reached a constant mass (tfinal). Weight percent moisture for samples at $t_1$ and $t_2$ and the moisture content reduction rate were calculated according to Eqs 2–4.

$$w_{t_1}\% = \frac{[(m_4 - m_1) - (m_6 - m_1)]}{(m_3 - m_1)} * 100\% \tag{2}$$

$$w_{t_2}\% = \frac{[(m_5 - m_1) - (m_6 - m_1)]}{(m_3 - m_1)} * 100\% \tag{3}$$

$$R = \frac{W_{t_1}\% - W_{t_2}\%}{|t_1 - t_2|} \tag{4}$$

In these expressions, $m_1$, $m_2$, $m_3$, $m_4$, $m_5$, and $m_6$ are masses of the beaker, added oily sludge, added OSM or chemical agent, and masses recorded at $t_1$, $t_2$ and tfinal, respectively. R is the moisture content reduction rate.

**Thermo Gravimetric Analysis (TGA).**   Approximately 10.0 mg samples of oily sludge were placed in the sample chamber (Synchronous Thermal Synthetic Analyzer, Model No. TGA/SDTA851e). The sample was held at 100˚C for 10 min, and the drying curve was then generated.

**Microscope analysis.**   The oily sludge microscope analysis was conducted using a Motic M150 SERIES microscope. (1) Oil sludge (100–500 mg) was placed on the slide and gently covered with a glass slide to observe the presence of water under the microscope. This was compared with the oil sludge sample with the added OSM, which was similarly examined. (2) The

existing forms of the leaching water in the oily sludge were determined by adding the OSM to 500.0 mg of oily sludge leachate. The oily sludge leachate was placed on the slide, gently covered with a glass slide, and the presence of water was observed under a microscope. Under the microscope, the emulsified water appeared as an opaque droplet with outer ring, while the free water was relatively uniform and transparent.

**Scanning electron microscope (SEM).** The samples were first dried to a constant weight at 80°C. A small amount of sample powder was attached to the copper platform, and then the sample was sprayed with gold to increase the conductivity of the sample, making the image clear and stable. The gold sprayed samples were placed in a vacuum system in the scanning electron microscope and were vacuumed and scanned. During the test, images were displayed by a computer system, and the results were recorded at resolutions of 10 μm and 2 μm, respectively.

## Results and discussion

### Preliminary screening of modified materials raw materials

The effect of single auxiliary materials on the moisture content of oily sludge was studied at a temperature of 10–15°C. **Table 1** shows the experimental results.

With the exception of $CaCl_2$ and $NaSiO_4$, when chemical reagents were added, the moisture content was lower compared to when no chemicals were added. This indicates that these agents positively impacted the natural drying of oily sludge (**Table 1**). However, adding only a single chemical reagent failed to achieve a moisture content of less than 50% within 5 days, and the added masses reached 10%. The greater the added amount of chemical reagents was, the higher the transport and disposal costs to the brick factory. The research goal was to achieve a shorter time, a lower moisture content, a lower cost, and a lower weight; the lower these values were, the better the outcomes were. Combined with modified materials design principles, it was necessary to screen the treatment agents for compounding effects.

### Determination of the OSM formulation by orthogonal experiment

Several indoor compounding experiments were conducted to assess the four types of chemical agents (Amino sulfonic acid, $Fe_2(SO_4)_3$, Portland cement, $MgCl_2$) listed in **Table 1**. The goal was to assess the best combination of OSM components. Orthogonal experimental methods were adopted to obtain the best ratio and reduce the experimental strength. To control the

**Table 1. Preliminary screening and evaluation results of modified materials.**

| Additive type | Name/ Chemical formula | Moisture content before administration | Moisture content after 5 days (%) |
|---|---|---|---|
| powder emulsifier | Sodium alpha-olefin Sulfonate | 71.06 | 59.76 |
| | Aminosulfonic acid | 71.06 | 58.84 |
| | sodium Dodecyl benzene sulfonate | 71.06 | 63.21 |
| oxidant | $FeCl_3$ | 71.06 | 61.44 |
| | $Fe_2(SO_4)_3$ | 71.06 | 58.31 |
| inorganic cementitious material | CaO | 71.06 | 53.23 |
| | Portland cement | 71.06 | 52.44 |
| other reagents | $CaCl_2$ | 71.06 | 66.84 |
| | $MgSO_4$ | 71.06 | 64.49 |
| | $NaSiO_4$ | 71.06 | 65.90 |
| | $MgCl_2$ | 71.06 | 63.69 |
| blank sample | / | 71.06 | 65.08 |

**Table 2. Orthogonal experimental factors and levels of OSM complex (%).**

| Level | Factor | | | |
|---|---|---|---|---|
| | Aminosulfonic acid (marked as A) | Fe2(SO4)3 (marked as B) | Portland cement (marked as C) | MgCl2 (marked as D) |
| 1 | 0.2 | 0.2 | 6 | 1 |
| 2 | 0.4 | 0.5 | 7 | 1.5 |
| 3 | 0.6 | 0.7 | 8 | 2 |

increase in the total weight caused by the increase of the modified materials, the total content of each component in the modified materials was experimentally controlled to be less than 10% larger compared to the oily sludge sample mass.

In this experiment, the four agents and three horizontal orthogonal tests were used to remix the OSM. The moisture content was assessed 5 days after adding the OSM. The moisture content and the price of a single reagent were used to develop an assessment index for the four treatments: Aminosulfonic acid (marked as A), $Fe_2(SO_4)_3$ (marked as B), Portland cement (marked as C), $MgCl_2$ (marked as D), A, B, C, and D included only one of each of the chemical agents listed in **Table 1**. Theoretical analysis and verification were used to determine the best ratio of the 4 chemical reagents. **Table 2** shows the determination of the factors in the orthogonal tests; A, B, C, and D were the 4 treatments; and 1, 2, and 3 indicated the treatment levels.

Based on the factors and levels above, we developed the $L^9(3^4)$ orthogonal table. The values in Table 4 were calculated from the experimental results in Table 3. Ki (i = 1,2,3) represents the sum of the test results corresponding to the horizontal number of i in any column. For example, when the level number of $K_1$ is 1 and the corresponding level number of factor A is 1, $K_1$ is equal to the sum of the results of the moisture content measurement data in each row when A is 1 in Table 3; ki = Ki/s, s is the number of occurrences of each level in any column; and R represents a range, R = Max (Ki) -min (Ki), or R = Max (ki)-min (ki). The larger R is, the more important the factor is, and the smaller the ki value is, the more it indicates that the factor i is the optimal choice in this column. The outcome shows that the optimal combination of moisture content after 5 days was $A_1B_2C_3D_3$ (Table 4). This means that treatment A was best at level 1, treatment B was best at level 2, and treatments C and D were best at level 3. Therefore, the best OSM mixture formulation, based on the mass of the sludge, as a percentage of mass was: 0.2% A, 0.5% B, 8% C, and 2% D. That is, the best OSM was an oily sludge modified material made from a mixture of 0.2% Aminosulfonic acid, 0.5% $Fe_2(SO_4)_3$, 8% Portland cement, and 2% $MgCl_2$.

**Table 3. Orthogonal experimental results of OSM recombination.**

| Number of experimental groups | Factor level numbers | | | | Moisture content (%) |
|---|---|---|---|---|---|
| | A | B | C | D | |
| 1 | 3 | 2 | 1 | 1 | 48.22 |
| 2 | 2 | 3 | 1 | 2 | 47.83 |
| 3 | 1 | 1 | 1 | 3 | 45.96 |
| 4 | 3 | 1 | 2 | 1 | 44.88 |
| 5 | 1 | 3 | 2 | 2 | 43.05 |
| 6 | 2 | 2 | 2 | 3 | 42.62 |
| 7 | 3 | 2 | 3 | 1 | 42.89 |
| 8 | 2 | 1 | 3 | 2 | 43.66 |
| 9 | 1 | 3 | 3 | 3 | 43.26 |

**Table 4. Data processing results table.**

| Number of experimental groups | | A | B | C | D |
|---|---|---|---|---|---|
| Moisture content after 5 days/% | K1 | 132.27 | 134.5 | 142.01 | 135.99 |
| | K2 | 134.11 | 133.73 | 130.55 | 134.54 |
| | K3 | 135.99 | 134.14 | 129.81 | 131.84 |
| | k1 | 44.09 | 44.83 | 47.34 | 45.33 |
| | k2 | 44.70 | 44.58 | 43.52 | 44.85 |
| | k3 | 45.33 | 44.71 | 43.27 | 43.95 |
| | R | 1.24 | 0.26 | 4.07 | 1.38 |
| | Excellent level | A1 | B2 | C3 | D3 |

### Effect of OSM on semi-drying of oily sludge

**Effect of OSM addition.**　A sample with OSM added was placed at rest for 120 h at 10–15˚C. Changes in the moisture content over time are shown in **Fig 2** (**S1 and S2 Tables in S1 File**). With the gradual increase in the OSM, the moisture content of oily sludge showed a significant downward trend. The larger the amount of added OSM was, the lower the final moisture content of the oily sludge was at the end of the experiment. The water evaporation rate of

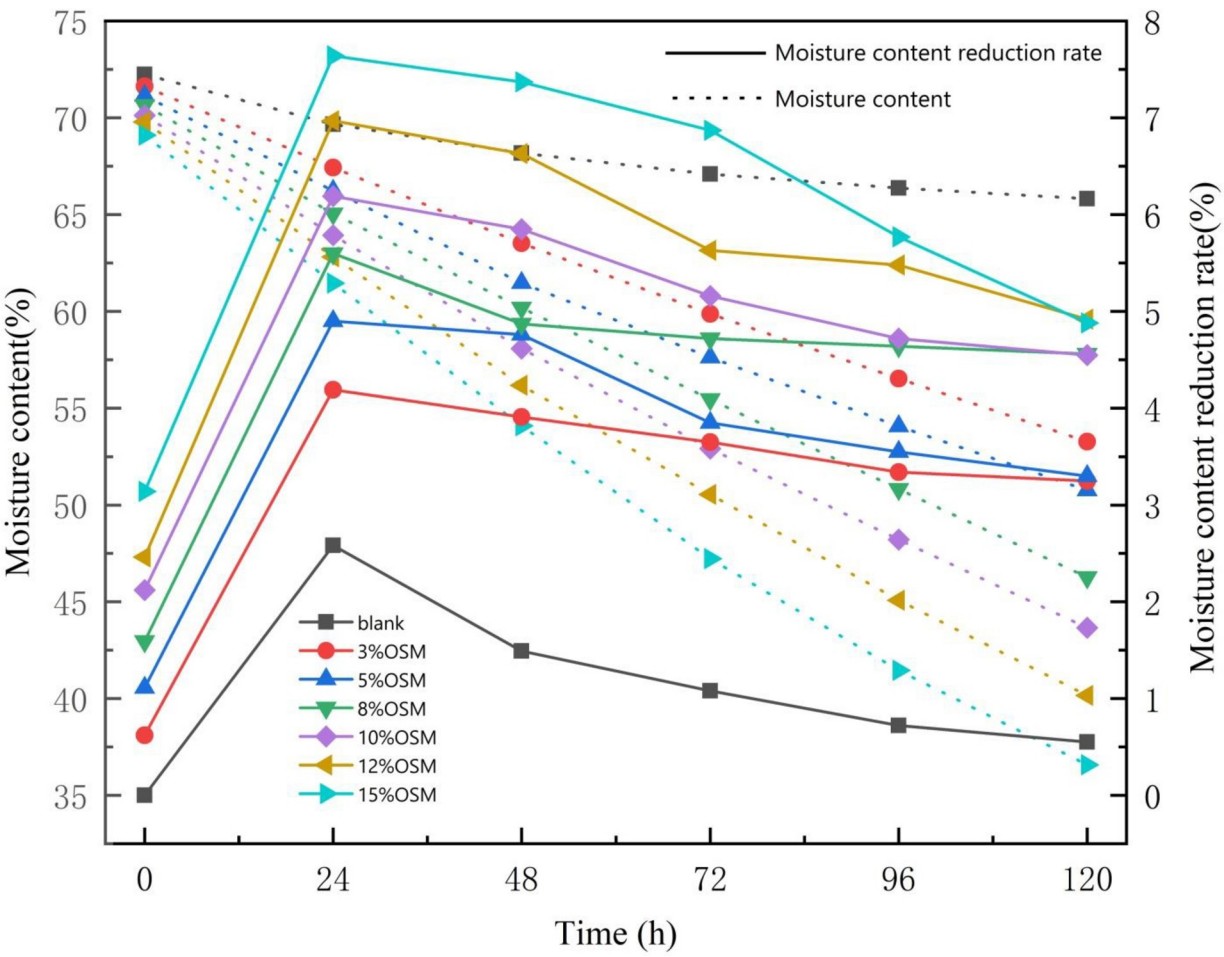

**Fig 2. The variation curve and reduction rate of moisture content after adding different levels of OSM.**

the oily sludge increased with OSM content (**Fig 2**). With the addition of the OSM and the solid-phase water absorption reaction, the moisture content of the oily sludge decreased. When the amount of added OSM increased from 3% to 15%, the moisture content of oily sludge decreased by 0.62–3.14% after being evenly mixed (0 h, starting time). There was an upward trend of moisture content reduction rate as the OSM mass increased. Twenty-four hours after the OSM was added, the moisture content reduction rate increased from 2.58% to 7.64%. The moisture content of the oily sludge dropped significantly faster compared to when it was initially added during the treatment period. The moisture content of the oily sludge with the added OSM was significantly higher compared to the blank sample. The OSM consumed less water during the pure physical adsorption of oily sludge. However, the evaporation rate of water in oily sludge was increased by the synergy with OSM.

The rate of the drop in water content declined over time. After 72 h, the value dropped more slowly. At this time point, moisture content dropped below 60% for all OSM treatments. The curves of moisture content reduction rate in **Fig 2** show two trends: fast moisture content reduction during the first 24 hours and rates slowly decreasing after 24 hours. These trends are consistent with the processes of evaporation and diffusion as described by Zhu et al., 2012 [20].

(1) Evaporation was possible due to the water vapor present on the surface of the oily sludge. The removal of water from the surface of the oily sludge occurred through water migration and evaporation. This was seen in the initially high moisture content reduction rate. This occurred in two main stages. First, the water vapor pressure became lower compared to the medium (gas). Second, due to the partial pressure of the water vapor, the moisture moved from the oily sludge surface to the medium.

(2) The diffusion process is a mass transfer process that is closely related to vaporization. When the moisture on the surface of the oily sludge evaporated, the humidity on the surface of the oily sludge was lower than the internal humidity of the material. This was seen in the final low moisture content reduction rate. A heated driver was needed to transfer the moisture from the interior to the surface. When the drying began, the mechanism driving the movement of the water was through capillaries in the pores of the walls of the solid, extending to the surface. This was conducted in an accelerated manner, reflecting a rapid decrease in water activity in the oily sludge. At the end of the drying process, there was a decrease in the enthalpy balance between the temperature of the air and the oily sludge. The water activity of the sludge reached equilibrium with the relative air humidity. This made water migration increasingly difficult [6].

The continuation and alternation of the above two processes essentially reflect the drying process. This is consistent with the theory that drying is complementary and parallel to surface vaporization and internal water diffusion.

**Effect of temperature.**   The moisture content curve of the oily sludge over the 120 h at different temperatures after adding 10% OSM is shown in **Fig 3** (**S3 Table in S1 File**). The water content of the oily sludge decreased significantly as the temperature increased. The evaporation and diffusion of water in oily sludge is considered a one-dimensional heat transfer process in porous media [21, 22]. When the elevated temperature reached the oily sludge surface, this increase in the surface temperature generated an increase in the mass fraction of free water at the surface and an increase of the evaporation velocity. The moisture on the surface of the oily sludge began to vaporize and evaporate. However, there were different sources of resistance to the diffusion of the wet points inside the porous medium.

As the natural drying process progressed, the rate at which the internal moisture of the oil sludge migrated to the surface became slower than the vaporization rate of the surface moisture. Some "dry zones" appeared, indicating a decrease in the actual evaporation surface area.

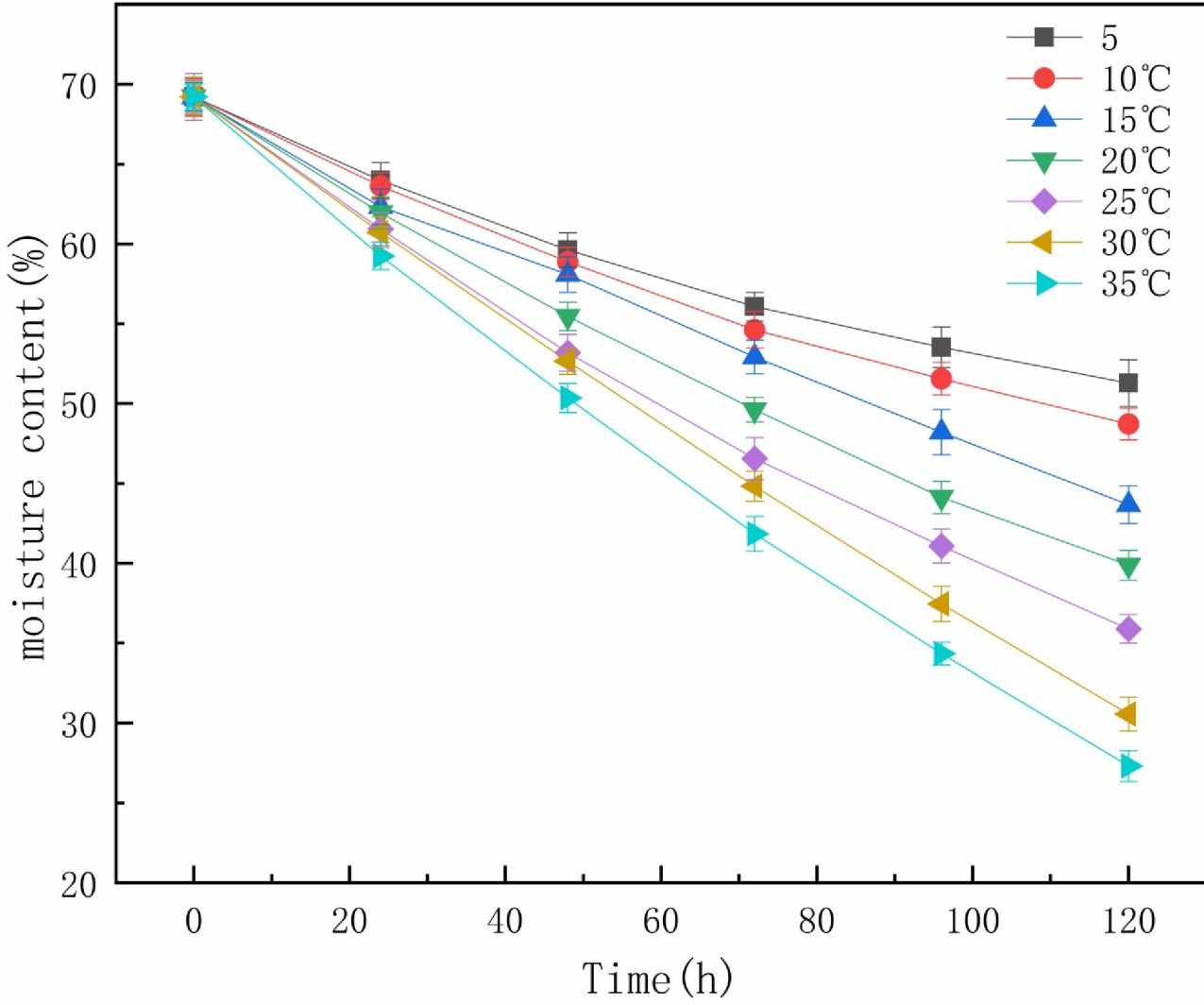

**Fig 3. Effect of temperature on the moisture content of oily sludge.**

Air and oily sludge convection heat transfer, which is the heat transfer of air from the surface of oil sludge to the interior of oil sludge, can achieve convective heat transfers between hot air and oil sludge. When the free water inside the oily sludge is heated, it gradually migrates to the surface and diffuses into the air. Increasing the temperature accelerated the heat transfer and increased the vaporization speed, speeding up the natural semi-drying of oily sludge.

If the oily sludge is required to be semi-dry within 120 h after adding 10% OSM, the temperature should be maintained at greater than 10˚C (**Fig 3**). In actual production, the semi-drying time of oily sludge can be shortened or prolonged according to the increase and decrease of ambient temperature at the same OSM dosage.

## Mechanism of OSM on natural semi-drying of oily sludge

**Conversion of existing forms of water in oily sludge.** *Natural evaporation curve of distilled water and emulsion.* The weight and rate of the natural evaporation of distilled water and oily sludge leachate were measured. The curve in **Fig 4** (**S4 and S5 Tables in S1 File**) shows

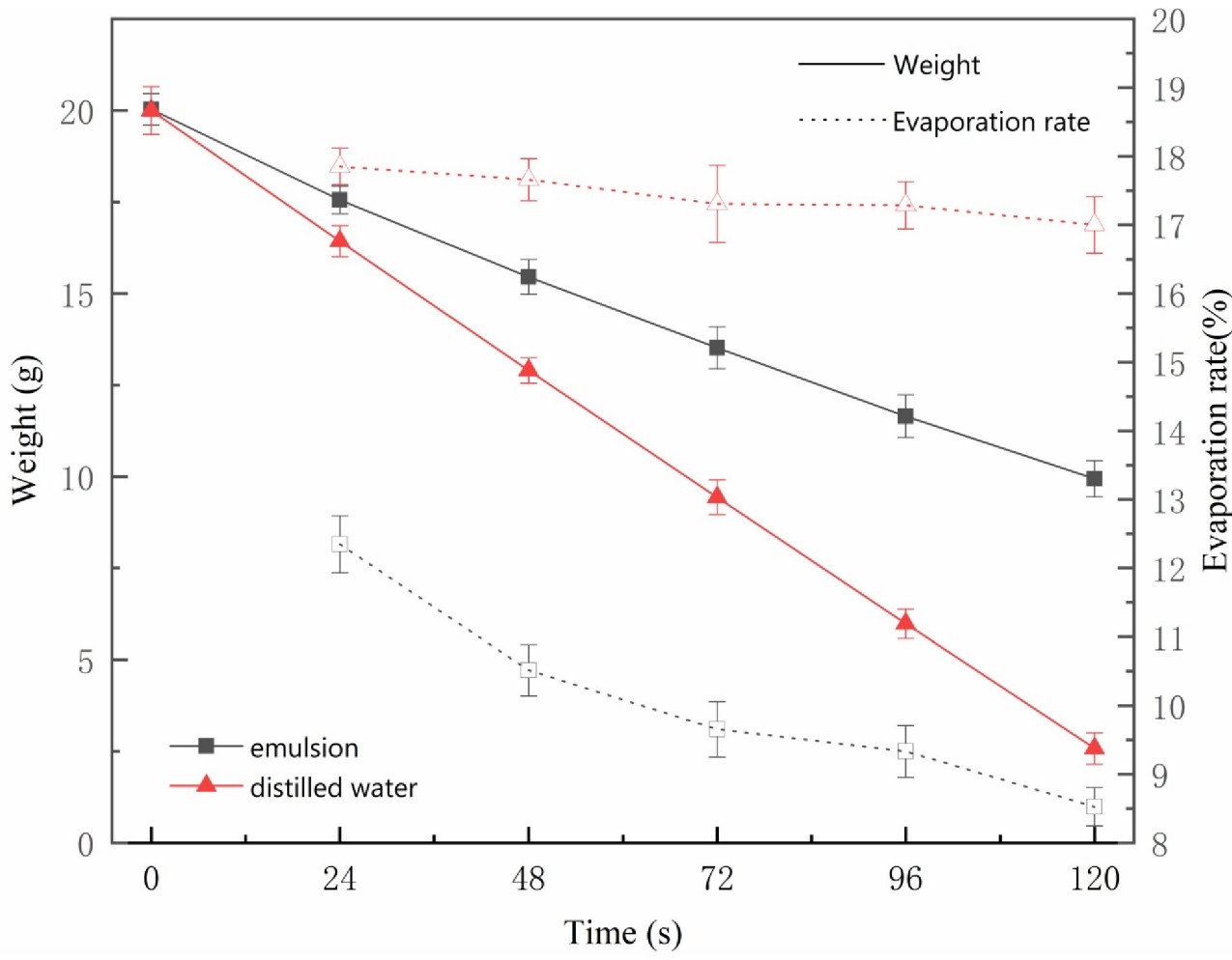

**Fig 4. Changes in evaporation of distilled water and emulsion.**

that the evaporation of distilled water was greater than the evaporation of the emulsion at the same drying duration and temperature (the evaporation area was also the same). This is because of the molecular membrane surface in the emulsion system [23]. As the centration of the membrane surface of the leachate increased, the molecular area decreased, the molecules moved closer to each other, the intermolecular force strengthened, the surface pressure increased, and the molecular membrane condensed to inhibit water evaporation.

From **Fig 4**, the drying speed of the water was slightly reduced. The change was not significant, but the drying rate of the leachate was reduced. When it began to dry, the desiccation speed of the emulsified water was lower compared to the distilled water. Over time, the moisture in the emulsion decreased. Although the chance of collision and rupture of the droplets increased, part of the free water was released. However, the coating thickness and area of the emulsion surface oil film increased, and the resistance generated by the surface oil film also increased. This led to a significant decrease in the drying rate of the emulsion. When the surface oil film was completely covered and the thickness reached a certain level [24], the oil film molecules were closely arranged. It could not be compressed again, and the drying speed of the emulsion stabilized at a lower speed.

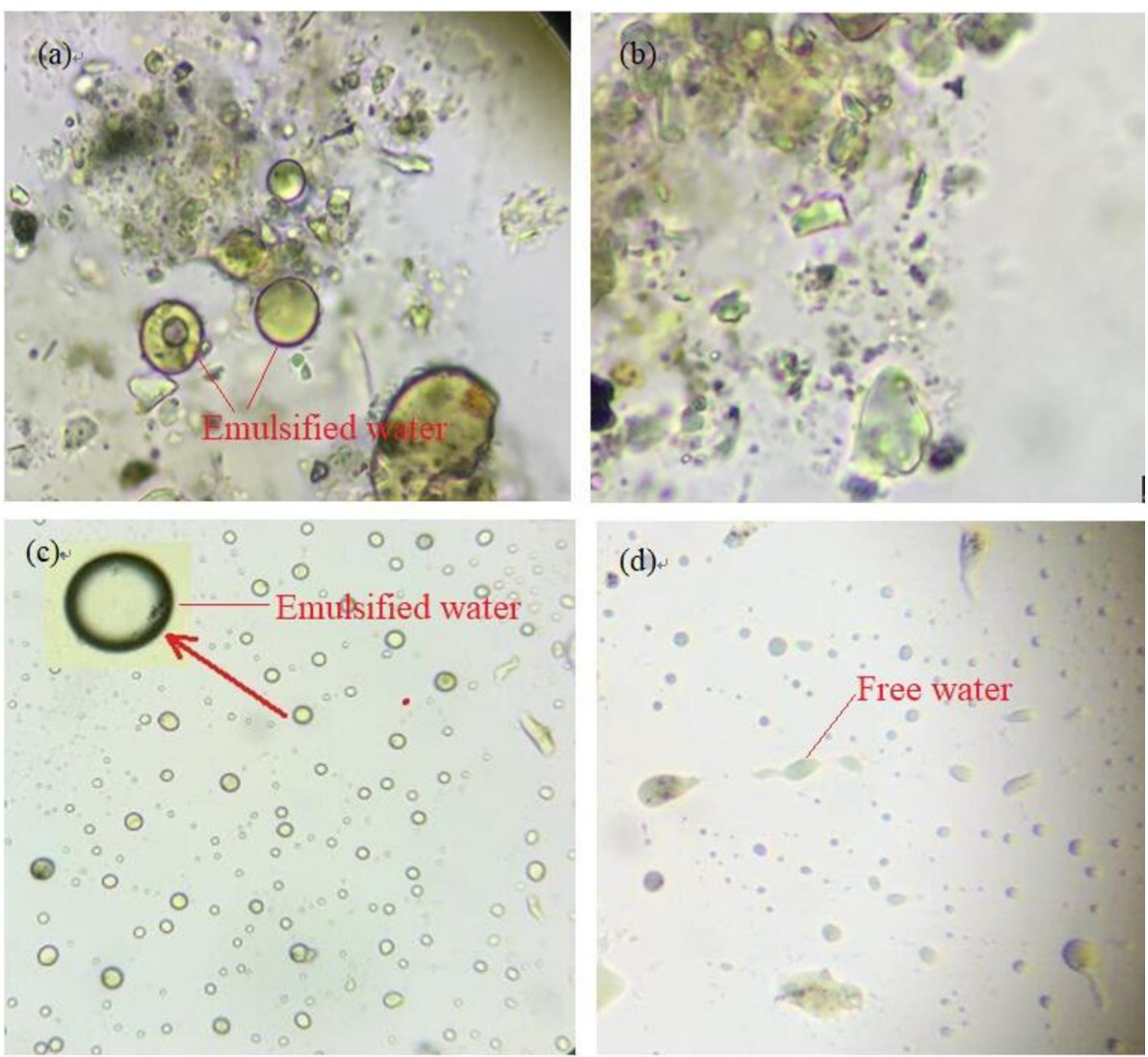

**Fig 5.** The status of the water in the crude oily sludge (a); the water in the oily sludge after adding OSM (b); the status of the water in the original oily sludge leachate (c); and the water in the oily sludge leaching solution after adding OSM (d).

*The impact of adding OSM*: *Before and after*. There was a large amount of water in the original oily sludge, and there was multiple emulsified water. **Fig 5** shows the status of the water in the crude oily sludge (a); the water in the oily sludge after adding OSM (b); the status of the water in the original oily sludge leachate (c); and the water in the oily sludge leaching solution after adding OSM (d). The surface of the oily mud particles was oil-friendly (**Fig 5A**). After the OSM was added, the emulsion was essentially destroyed and the oil layer on the oil surface disappeared (**Fig 5B**). There was a significant change in the presence of water in the oily sludge leachate after adding OSM. This is consistent with the observations shown in **Fig 5C and 5D**.

*Surface tension measurement result*. Surface tension test results showed that the surface of the original oily sludge leaching solution had a surface tension value of 33.89 N/m (**S1A Fig in S1 File**). Once the OSM was added, the surface of the oily sludge leachate had a surface tension value of 43.97 N/m (**S1B Fig in S1 File**). The results show there was a change in the presence of water in the oily sludge: hydrophilicity increased, and the lipophilicity decreased.

**Mechanism of the existing form of water in OSM transformation oil sludge.** Two mechanisms caused the OSM to change the type of water in the oily sludge. First, the OSM's inorganic cementitious materials and other reagents contain many electrolytes. These compress and reduce the diffusion of electric double layers on the oil bead surfaces. The reduction in the surface charge of oil beads increases the opportunity for fuel beads to be added to the collision [25]. Second, the solid surfactant in the OSM changed the hydrophilic and oil-friendly equilibrium value of the water in the oily sludge. Additionally, the surfactant was adsorbed on the moist clay particle surfaces to change the wettability and destroy the oily emulsion [26]. Finally, the negatively charged species in the OSM accelerated the electrical polarity of the aqueous oil and increased the probability of damage to the emulsion [27].

**TGA analysis result.** The mass of oily sludge sample was reduced by 69% after drying for 10 min at 100˚C (**Fig 6**). The evaporation rate decreased in turn. However, **Fig 6** shows the difference in the weight reduction curve per second for the oily sludge samples. From the beginning of the experiment to point a, as the temperature rapidly rose to 100˚C, the rate of weight loss gradually increased. After the temperature reached 100˚C, the rate of weight loss increased sharply (a to b stage): as the free water decreased, the weight decreased. The rate was reduced between b and c as remaining free water evaporated. The water in the form of emulsified water in the oily sludge was demulsified and the free water increased over a short time, leading to an increase in the rate of mass loss (c to d phase). As the emulsified water decreased, the heat was

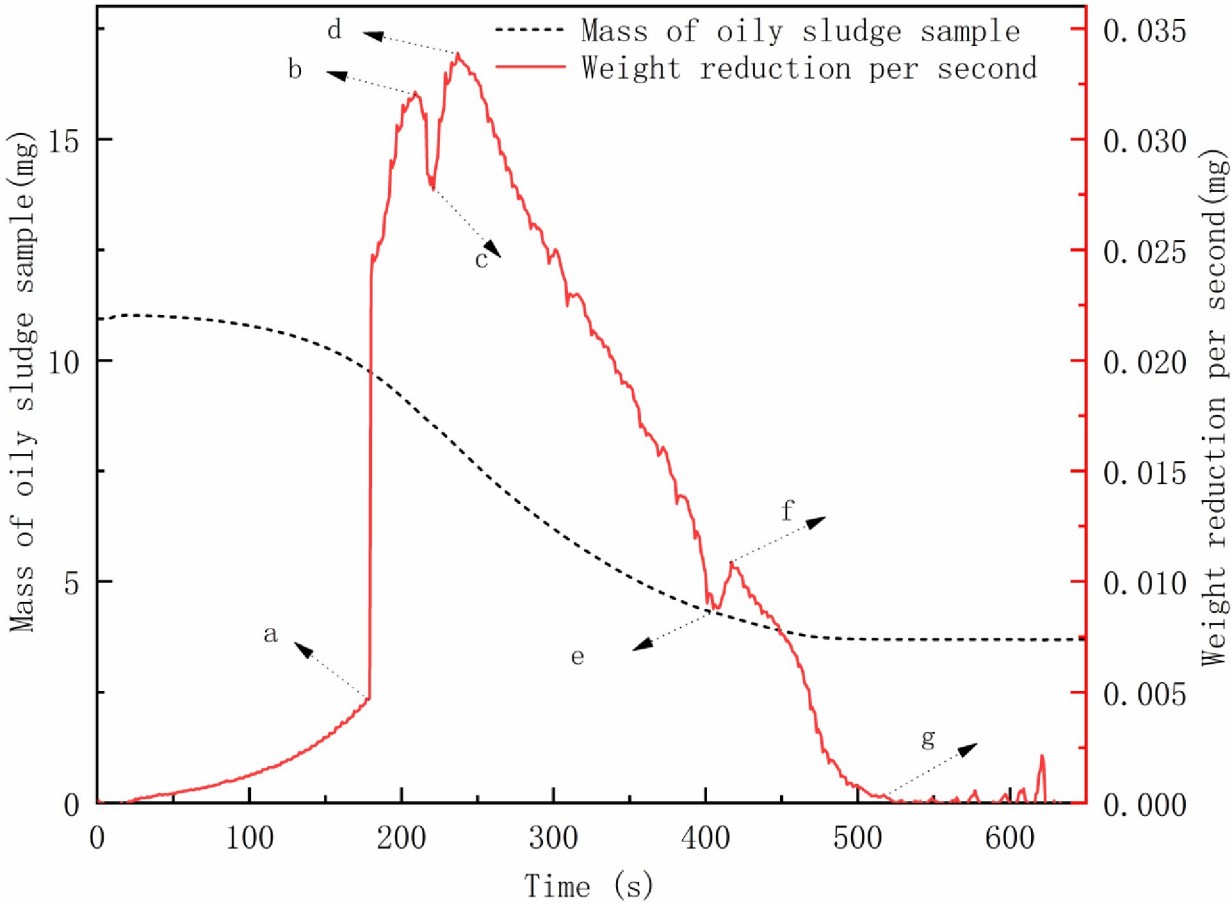

**Fig 6.** Evaporation curve of oily sludge sample (a, b, c, d, e, f, g are the typical inflection points for the evaporation rate).

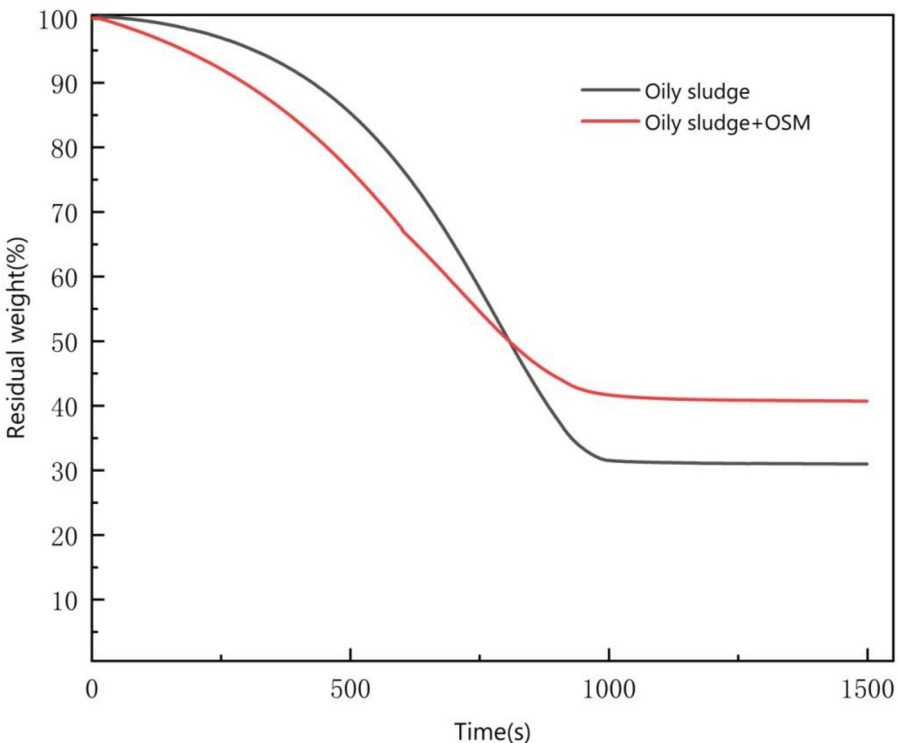

**Fig 7. Thermo gravimetric analysis (TGA) curves.**

transferred from the surface of the sludge to the interior, and the bound water between internal pores of oily sludge particles inside the oily sludge was transferred to the surface through the pores. However, the internal velocity was affected, and the transfer speed was significantly smaller than the evaporation rate. Therefore, the rate of weight reduction was gradually reduced (c to e phase).

When point e was reached, the free water and oil-bearing sludge particles surface bound water were significantly reduced, and the weight reduction speed was minimized. At this point, the water in the state of the emulsion inside the oily sludge broke for a second time, and the interior free water of the oil sludge quickly increased. This resulted in increased evaporation (e to f stage), as the internal moisture content of the oily sludge was further reduced, and the evaporation rate decreased, reaching point g. The water, in the form of bound water between internal pores of oily sludge particles, and all free water evaporated. The weight reduction trended towards 0 mg/s, and the evaporation process was essentially completed.

The oily sludge TGA curves are shown in **Fig 7**. They show that after adding 10% OSM, there was a significant reduction in the weight of the oily sludge at the beginning of heating. Over time, the weight reduction rate decreased, stabilizing after about 1000 s. A similar length of time was required for the non-amended oily sludge drying to stabilize at a constant weight. The experimental process and the TGA curves show the stages to the drying process [28].

1. At the start of the experiment, the heating temperature rose from 30˚C to 50˚C at a rate of 5˚C per min. When the surface of oily sludge was heated, the free water on the surface began to evaporate. After oily sludge sample was added to the OSM, the oily sludge sample first reacted with free water due to water absorption from the powdered material. The water in the emulsion form was gradually converted into free water. The emulsified water transformed into free water form. The reaction rate became faster.

2. During the rapid evaporation phase, the heating temperature rose from 50°C to 80°C, the temperature accelerated water molecule activity, the emulsion in the oily sludge sample gradually broke down, and there was an increase in the speed at which the water inside the oil sludge particles migrated to the surface. The reaction rate was significantly higher compared to the previous stage. After the oily sludge sample with the OSM was heated, the migration of the free water to the surface was significantly faster than the migration of water present in the emulsion form. At the same time, the hydration diffusion layer of inorganic cementitious materials rapidly lost water, the porosity increased, and the speed of water migration accelerated.

3. In the deceleration evaporation stage, the heating temperature rose from 80°C to 100°C. At that time, the free water, the water in the emulsion, and the bound water were almost evaporated, and there was a reduction in the difference between the moisture content of the water molecules and the surface of the oily sludge. The water molecules had greater difficulty in migrating to the surface [29], and the evaporation rate of free water and emulsified water decreased. However, due to the presence of the inorganic gel material as a skeleton structure, there was a reduction of capillary pores during the oily sludge drying process.

4. At the end of the reaction, except for a small amount of internal crystalline water in the oily sludge, the water had essentially evaporated. As time passed, the reduction in the weight of the oily sludge almost stopped, and the weight remained constant.

**Hydrochemical damage of inorganic cementing material emulsion surface film.** After the OSM was added to the oily sludge, inorganic cementitious particles (mainly cement) in the OSM reacted rapidly with the free water in the oily sludge. Within a few minutes, a cemented film layer formed on the particle surfaces, and crystals gradually precipitated [30, 31]. This

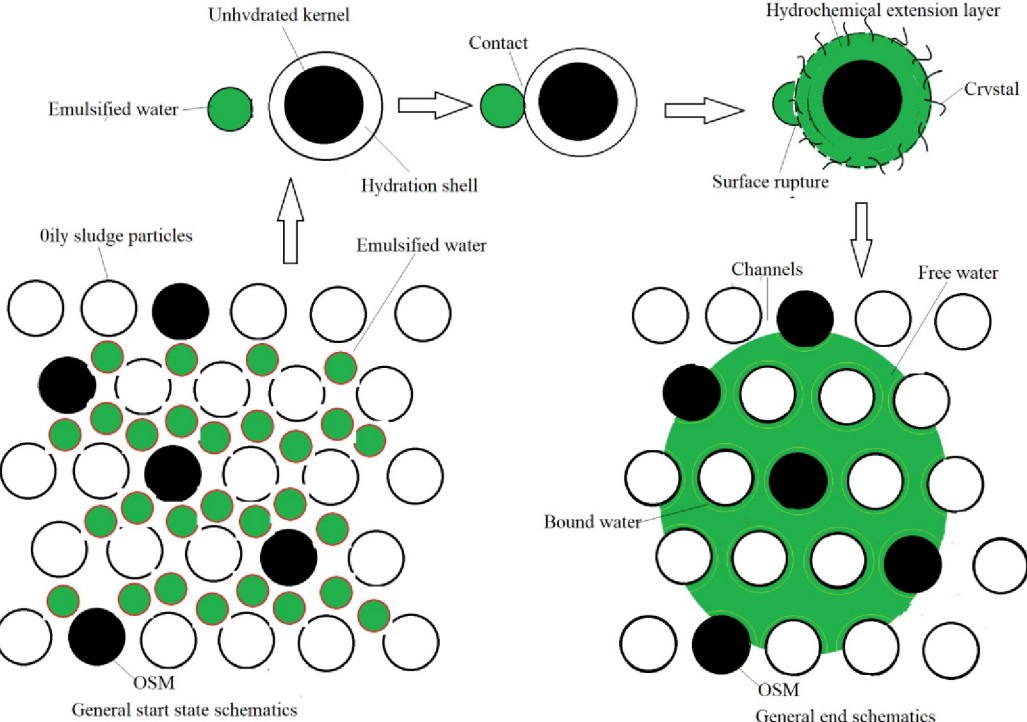

**Fig 8. A schematic diagram of damage to emulsion surface film during hydration of inorganic cementing materials.**

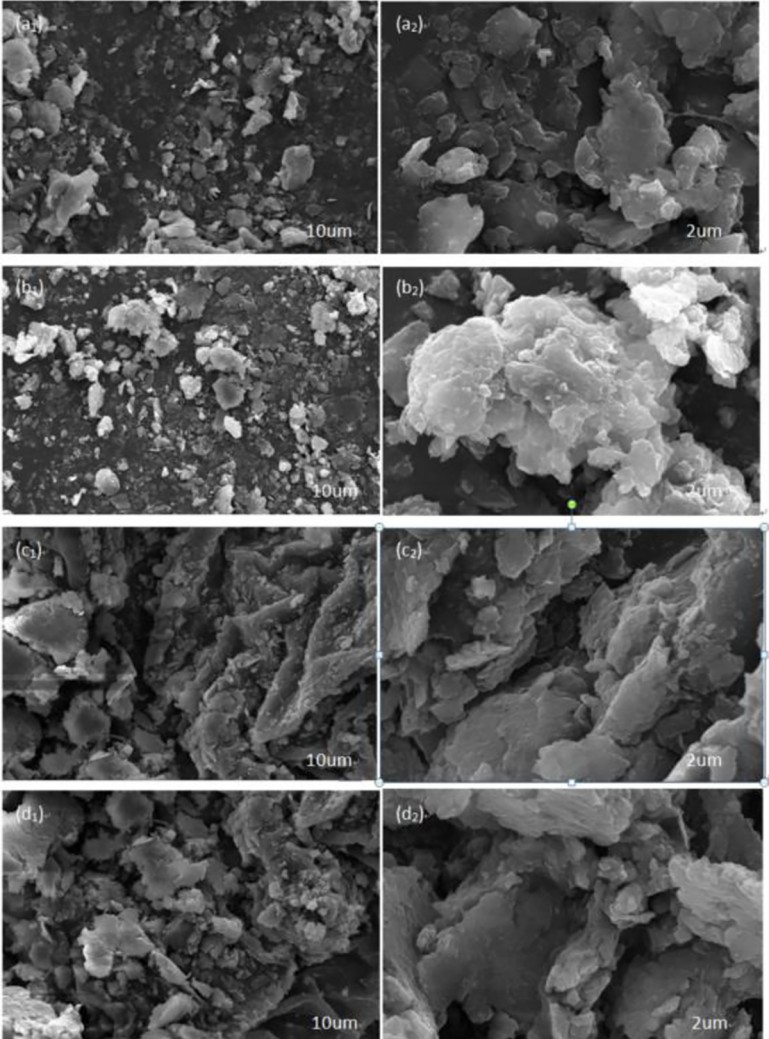

**Fig 9.** SEM photographs of oily sludge: Oily sludge samples ($a_1$,$a_2$); Natural dry oily sludge samples added to CaO ($b_1$, $b_2$); Natural dry oily sludge samples added to Cement ($c_1$,$c_2$); Natural dry oily sludge samples added to OSM ($d_1$,$d_2$).

formed a hydrated diffusion layer. During the formation of the gelatinous membrane and diffusion layer [32], large water emulsions in the pores of the oily sludge contacted the inorganic gelatinous particles. The emulsion surface film layer was broken through adsorption and extrusion. Emulsified water was converted to free water (**Fig 8**). The free water particles were smaller than the water in the emulsion form, making it easier for the water to migrate from the interior of the oily sludge to the surface through capillary holes. This accelerated the natural drying rate.

The channels that allowed the vapor to escape were critical for water removal [33]. **Fig 9** shows the SEM photographs of the oily sludge samples before and after drying. This allows for a comparison of the SEM images of oil sludge blank samples and the oil sludge samples with CaO, cement, and OSM.

During the natural drying of the oily sludge, even when the pore water has been converted into free water, sufficient pore passages are needed for the water to migrate from the inside to the surface [34]. **Fig 9A** shows the dense surface of the dry-oily sludge blank sample. It is

possible that during the drying of the oil-containing sludge, as the free water evaporated, the volume of oil-containing sludge shrunk toward the center after the oil-bearing sludge particles surface-bound water evaporated. This formed a dense surface structure with limited routes for water migration. A large amount of non-free water was trapped, causing the natural semi-drying process to require more time and energy. **Fig 9B–9D** shows that the oily sludge increased the loose flocculation of many small particles connected, and the structure of oily sludge loosened. The skeletal structure [35] significantly increased the porosity of the oily sludge for water migration. The CaO, cement, and OSM were each added to change the surface characteristics of oily sludge, increasing the porosity of the oily sludge from the inside to the surface. The SEM characterization indicated that the most important effect of adding OSM was the creation of a loose surface structure on the oily sludge. The flocculation was small and dispersed, supporting the conversion of the existing form of water in the oily sludge into free water. This indicated that the structure of the sludge solid was remodeled into a more stable form. The migration of the skeleton's porous channels to the surface, and the resulting rigid structure, benefitted the release of the bound water [36].

## Conclusions

This study found that an OSM mixture reduced the natural semi-drying time of oily sludge and the resulting disposal cost. The OSM was composed of powder emulsifier, oxidant, inorganic cementitious material, and other reagents. At environmental temperatures exceeding 10˚C, the OSM significantly shortened the natural semi-drying time of the oily sludge from 15 days to 5 days. Moreover, the natural evaporation rate of free water with the added OSM was higher compared to the evaporation rate of the emulsion and bound water. Adding OSM changes the existing form of water in oily sludge, causes the emulsion to break, reduces surface tension, reduces bound water, and releases free water. At the same time, it increases the internal micro-pores of oily sludge, facilitating water migration from the core of oily sludge to the surface and air, and speeding up the natural semi-drying of the oily sludge. In the process of natural drying, other factors impact the process, including humidity, air pressure, and the thickness and compaction of the oily sludge piles. These impacts deserve further study.

## Supporting information

**S1 File.**
(RAR)

## Author Contributions

**Data curation:** Meng Zhu.

**Methodology:** Maoqi Liao.

**Writing – original draft:** Maoren Wang.

**Writing – review & editing:** Yucheng Liu, Mingyan Chen.

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
