## [Decision Letter · Decision Letter 0]

17 Sep 2020

PONE-D-20-21667

Promoting the natural semi-drying of oily sludge by changing the form of water

PLOS ONE

Dear Dr. Liu,

Thank you for submitting your manuscript to PLOS ONE. After careful consideration, we feel that it has merit but does not fully meet PLOS ONE’s publication criteria as it currently stands. Therefore, we invite you to submit a revised version of the manuscript that addresses the points raised during the review process.

We look forward to receiving your revised manuscript.

Kind regards,

David Rider

Academic Editor

PLOS ONE

Journal Requirements:

2. Please amend either the title on the online submission form (via Edit Submission) or the title in the manuscript so that they are identical.

Reviewers' comments:

Reviewer's Responses to Questions

**Comments to the Author**

1. Is the manuscript technically sound, and do the data support the conclusions?

Reviewer #1: Partly

2. Has the statistical analysis been performed appropriately and rigorously? 

Reviewer #1: No

3. Have the authors made all data underlying the findings in their manuscript fully available?

Reviewer #1: Yes

4. Is the manuscript presented in an intelligible fashion and written in standard English?

Reviewer #1: Yes

5. Review Comments to the Author

Reviewer #1: Summary

This research addresses the issue of disposal of oily sludge, a common waste of the petroleum industry. These residues contain large amounts of water, thus increasing the volume and mass of material for disposal. Reducing the water content reduces the amount of waste and thus the economic burden of its disposal.

Authors use an orthogonal experimental method to determine the ideal mixture of chemical reagents (emulsifiers, oxidants, cementitious materials and “other reagents”) to use as an amendment to oily sludge to reduce drying times. The authors then carry out experiments to examine how temperature, exposure time and amendment concentration impact oily sludge drying time.

They show that chemical amendments can dramatically decrease the time needed for these wastes to reach a semi-dry (40-50% moisture content) state and thereby reduce the economic burden of oily sludge disposal. The novel aspect of this research is the application of chemicals to aid in reaching a semi-dry state. Research to date has focused primarily on physical (e.g. heat) methods of drying oily sludge.

Strengths: This manuscript provides a novel method of speeding up the drying processes of industrial oily sludge. The authors use an orthogonal experimental design and an efficient set of experiments to determine the most appropriate amendments to facilitate drying. Through this work they are able to decrease the semi-drying time of oil sludge by 2/3. Generally the manuscript is well written and has been edited.

Weakness: While generally a solid study on a novel method of handling oily sludge authors overstate the contribution of their work to our mechanistic understanding of the drying processes. Additionally, the study lacks statistical rigor requiring cautious interpretation of results.

Given the overall value of the research conducted, I recommend this manuscript for publication once appropriate revisions have addressed the concerns and editorial suggestions elaborated below.

Specific Areas for Improvement

Major

While generally a solid study and well written, there are several areas that I authors should address. First, the manuscript overstates its contribution to the mechanistic process of drying (e.g. lines 217 – 235, 258-270, 317-335, 343-371). While molecular-level mechanisms of drying discussed in the text are supported by the data presented, I do not think the data from this study provides direct evidence of these mechanisms. I suggest that reframing the discussion of mechanisms as possible explanations, consistent with observed data would be appropriate and more accurate as the mechanisms themselves are not explored in detail by the research performed. This should also involve the addition of relevant references to these sections of the text.

Second, a lack of statistical analysis of the data is also noted. The authors provide no information on sample replicates performed or calculation of errors, yet throughout the manuscript observed changes are noted as either significant or not significant (e.g. lines 202, 209, 239). If multiple experiments were carried out and errors on measurements are known, these values must be reported, especially if, as I suspect, variability in moisture content is high among different samples. Percent moisture values measured by the authors span less than 10% variability across these 11 treatments (and blank). Having some idea of the variability on these measurements would make it easier to accept the results of this study and allow greater understanding of the drying processes. While replicates and thorough statistical analyses are not always common practice in the field at this point, analysis of at least one sample in triplicate to provide some idea of variability would be extremely valuable. If available, error bars should be added to figures to aid the reader in accurate interpretation of the data. If authors do not have error data for their experiments and are not able to obtain this data, language of the manuscript must be revised and it should be clearly stated that each experiment was only performed once.

Minor

Third, authors should address the application of these experiments to larger-scale industrial processes that they discuss during the introduction. The experiments were carried out on very small sample sizes (10 mg), how do they anticipate their result scaling up? Is this research relevant to larger scale drying where surface area/volume ratios will be dramatically different? What recommendations do authors have for scaling for industrial application? What problems do they foresee?

Fourth, authors should briefly describe why modified materials were selected. I suggest elaborating on the discussion of the various OSM materials used (lines 300-310), adding appropriate references and moving it forward in the manuscript (to the introduction/Methods: Experiment section). It would be useful for the reader to understand why the various amendments chosen would be expected to aid in the drying processes prior to the OSM selection and experimental results.

Finally, authors must clean up language and clarify to reader the different forms of water discussed in the manuscript. This should include a clear description of each term used and consistent use of this terminology throughout the manuscript. There are currently over ten different terms used throughout the manuscript. (currently in use: free water, gap water, surface water, combined water, internal free water, pore water, absorbed water, combined water, water in emulsion, interstitial water, free emulsion…)

While the English of the manuscript is generally good there are still numerous typographical errors and areas that could use clarification. In addition to addressing these errors I encourage authors to complete a thorough read through of the paper to ensure their intended meanings have not been changed through the editing process. While not complete, below is a list of some specific suggestions and questions I noted as I read through the manuscript.

Notes on Figures and Tables

General: (1) Figure captions and Table headings could be more informative.

(2) Formatting for all tables could be improved. Values should be vertically centered and headings bolded.

Figure 3: Suggest authors graph this figure as % moisture vs Time. Current format is difficult to read and is inconsistent with how figure is discussed in the text. (Note: units on x-axis are currently incorrect)

Figure 6: Figure does not add to text or aid in data interpretation. Delete?

Figure 7: Interesting figure, currently not labeled sequentially (current: a, b, d, c …) which is confusing. Units for weight reduction per second should be listed. Y-axis (moisture content of oily sludge sample) is mislabeled.

Figure 8: define TGA in caption.

Figure 9: Suggest flipping this figure so transition process is shown below general start/end schematics to help reader. There are no clear structural changes between the initial and final schematics while text (lines 399 – 401) discusses a change in porosity. Suggest adjusting schematic to illustrate this change in porosity. Unclear where in the figure “free water” is shown. “thin holes” in final schematic discussed as “channels” in the text.

Table 1: remove “number” column unless used elsewhere.

Tables 2 and 3. Unless germane to the analysis/outcome I would suggest authors reorder tables so that they read in order A, B, C, and D as discussed in the text.

Table 3: no explanation or discussion of bottom portion of table (R/K values). If unnecessary should be removed. If kept should be discussed in text.

Line-by-Line notes:

- Line 25: “OSM increased…” increased how? by mass? Volume?

- Line 43: add “e.g.” to description – not all oily sludge has the same oil/water content, numbers provided are only one example of the composition of an oily sludge.

- Line 52: move/edit “These approaches also consume significant energy.” to line 54, before sentence on solar dryers.

- Line 57: delete “continuously” (not necessary)

- Line 57: “the analysis above” – There is no analysis above, and no discussion of natural wind drying. Statement is misleading. How wind better than solar? wind also varies.

- Line 71: “this leads to the need” … - no it does not, but it would be desirable to have a shorter drying period there is no evidence provided for a specific need of a drying time in fewer than 5 days.

- Line 76: edit – “wide source” � “widely available source”

- Line 82: need references – what studies?

- Line 114: “m,g” very confusing. I believe g is the units, unsure of m, mass? clarify

- Line 117: is it true that the material with the lowest moisture content was selected? Text states moisture content varied from 68-72% (line 102), table 1 says 71%

- Line 120: “Distilled water was used…” as a reference material?

- Lines 122-137: Wording could be simplified to facilitate understanding. A labeled schematic identifying each "m" would be useful for deciphering what each of these masses refers to. Supplemental figure?

- Line 137: Equation 4 does not provide a rate as is stated in the text.

- Line 140: define TGA

- Line 146: unclear: 100-500 mg is this the mass of the water or the sludge sample?

- Line 150: define SEM

- Line 154: SEM experiments = images collected?

- Line 155: use correct symbol for micro

- Line 170: use more appropriate word

- Line 175: “component of OSM” � “combination of OSM components”?

- Line 195: unclear as to if this formulation was used for all remaining experiments. If so add a sentence stating as much.

- Line 197: delete “adding”

- Line 198: delete “Effect of OSM Addition.”

- Lines 201-213: Authors suggest significant differences between experimental treatments but do not provide statistics to back up this claim.

- Line 208: There was an upward trend in what?

- Line 209-210: Data not shown – clarify for reader that this data is not presented

- Line 210-211: Moisture content of what was higher compared to blank?

- Line 215: edit - “dropped below 60% for all OSM treatments”

- Line 215-216: How do the curves indicate to authors which processes are responsible? This statement appears to be a reference to Zhu et al. I am not sure authors intend to reference the actual shape of their curves. How are points 1 and 2 visually seen in these curves. This is an example of where authors appear to discuss explanations of data as a direct product of the research performed. Strongly suggest authors take a close look at this section and edit for clarity.

- Line 234: moisture content dropped below 60% for the OSM amended treatments only. Clarify.

- Line 250-251: Confusing. Edit for clarity

- Lines 254-256: Confusing. Edit for clarity.

- Line 255: edit – “should be controlled at no less than…” �”should be controlled to within at least…”. Unclear where this conclusion is coming from.

- Line 262: past tense: “evaporation area was…”

- Line 264: Surface concentration of what?

- Section starting at 257: Generally section should be copy-edited for clarity. This study provides data in support of mechanisms but does not specifically determine the mechanisms itself. Additionally numerous unsubstantiated uses of “significant”. See weaknesses discussed above.

- Lines 268-269: These two sentences seem to directly contradict eachother. I am unclear as to the authors intent. For the emulsion or for the water? Rewrite for clarity

- Line 284 – 289: How is this data shown in the images?

- Line 296:”hydrophilicity increased and the hydrophilicity decreased”

- Line 297-298: Confusing. Edit for clarity.

- Section starting at line 300: Add references. Suggest relocating to Experiment or Introduction section.

- Line 312: raw data should be removed.

- Line 327/331/333: Continue guiding reader. “.. reduction was gradually reduced (c to e phase).” etc

- Line 338: “stabilizing after ~1000 s”

- Line 343-371: Confusing / feels redundant with previous section. One discusses data where no OSM was added the other with OSM addition – distinction is not clear in text and adds to confusion. Addition of the “four stages” of the drying process to the relevant figure (figure 8?) would be extremely helpful. Additionally, use of multiple terms to describe water forms adds to confusion. Simplify and be consistent

- Line 391: Delete “f”

- Line 395: edit – “This dense …closed space” � “Forming a dense surface structure with limited routes for water migration”?

- Line 397: “oily sludge amendments increased”?

- Line 408: I believe authors used an OSM mixture, not compound for their experiments (as noted above this should be clarified in the methods section)

- Line 410: missing period. “…disposal cost. The environmental …”

- Line 411: Again, unsure where this conclusion has come from, figure 3? Lines 254-255. Clarify.

- Line 475/486: references 18 and 22 are missing year information

- References: generally inconsistent formatting of dates across references

6. PLOS authors have the option to publish the peer review history of their article (what does this mean?). If published, this will include your full peer review and any attached files.

Reviewer #1: No

---

## [Author Response · Author response to Decision Letter 0]

13 Oct 2020

Responds to the reviewers’ comments: 

Reviewer #1: 

Weakness: While generally a solid study on a novel method of handling oily sludge authors overstate the contribution of their work to our mechanistic understanding of the drying processes. Additionally,the study lacks statistical rigor requiring cautious interpretation of results.

Given the overall value of the research conducted, I recommend this manuscript for publication once appropriate revisions have addressed the concerns and editorial suggestions elaborated below.

Response:

Thanks for your comments. Your suggestion is greatly appreciated. Through your comment, I deleted the text “significant effect”, “significant improvement” with exaggerated words, as well as possible objective description of the experimental data and conclusions.

Reviewer #2: 

Second, a lack of statistical analysis of the data is also noted. The authors provide no information on sample replicates performed or calculation of errors, yet throughout the manuscript observed changes are noted as either significant or not significant (e.g. lines 202, 209, 239). If multiple experiments were carried out and errors on measurements are known, these values must be reported, especially if, as I suspect, variability in moisture content is high among different samples. Percent moisture values measured by the authors span less than 10% variability across these 11 treatments (and blank). Having some idea of the variability on these measurements would make it easier to accept the results of this study and allow greater understanding of the drying processes. While replicates and thorough statistical analyses are not always common practice in the field at this point, analysis of at least one sample in triplicate to provide some idea of variability would be extremely valuable. If available, error bars should be added to figures to aid the reader in accurate interpretation of the data. If authors do not have error data for their experiments and are not able to obtain this data, language of the manuscript must be revised and it should be clearly stated that each experiment was only performed once.

Reply:

Thanks for your comments. Your suggestion is greatly appreciated. Through your comment, read the corresponding location, you do point to the existence of these problems, has been modified.

I'm sorry for our negligence , I have repeated the experiment and will do an error analysis. See figure2-3 for details.

Really helps readers understand the charts and data in this article, and it also helps the reliability of the data in this article. Having some idea of the variability on these measurements would make it easier to accept the results of this study and allow greater understanding of the drying processes.I have modified Figure 3 and Figure 4 and added error bars. Figure 2 has a total of 14 curves with dual coordinates due to more data points. There are more data lines. Error bars are not conducive to the observation of the graph, and the curves in the graph mainly show As the proportion of OSM increases, the moisture content of the sample and the decreasing rate of the moisture content change, so there is no increase in error bars.

Delete "more significantly" on page 202 and "significantly" on page 204, and amend page 202 to read "lower moisture content". Delete " significantly" on page 209 and page 239. Error bars had been added to figures.

Reviewer #3: 

Third, authors should address the application of these experiments to larger-scale industrial processes that they discuss during the introduction. The experiments were carried out on very small sample sizes (10 mg), how do they anticipate their result scaling up? Is this research relevant to larger scale drying where surface area/volume ratios will be dramatically different? What recommendations do authors have for scaling for industrial application? What problems do they foresee?

Reply:

Thank you for bringing up one of the main purposes of this study, which was to apply the results of the experiment to larger-scale industrial processes in which oil sludge containing 65%-80% water was dried under natural conditions and reduced to 50% water in a short period of time, to solve practical problems in production. The oil sludge samples in TGA were small sample sizes (about 10mg) , and in the other experiments, the oil sludge samples are 50.00 g, of course, compared with the industrial processes, these samples were very small sample sizes. However, the mechanism and influencing factors of the spontaneous semi-drying of oily sludge in industrial process were basically the same as that in this study, by adding OSM, the form of water in oily sludge can be changed and the water contained in the emulsion can be released. In the industrial processes, it is suggested that further experiments should be carried out on the stacking height and dumping frequency of oily sludge to find the best value for accelerating the natural semi-drying rate of oily sludge and to consider the best economic benefit scheme.Yes, this research relevant to larger scale drying where surface area/volume ratios will be dramatically different.

Reviewer #4: 

Fourth, authors should briefly describe why modified materials were selected. I suggest elaborating on the discussion of the various OSM materials used (lines 300-310), adding appropriate references and moving it forward in the manuscript (to the introduction/Methods: Experiment section). It would be useful for the reader to understand why the various amendments chosen would be expected to aid in the drying processes prior to the OSM selection and experimental results.

Reply:

Thanks for your comments. Your suggestion is greatly appreciated. The natural evaporation and diffusion rate of water in oily sludge were affected by the external environment and internal physical and chemical characteristics. The former was mostly the natural condition which can’t be changed, and the latter was the form of water, the interfacial tension of oil-water-solid three-phase, the migration and diffusion resistance of water could be changed by the action of chemical reagents according to the mechanism of sludge dewatering (Combined effects of Fenton peroxidation and CaO conditioning on sewage sludge thermal drying）. Therefore, it was predicted that the physical and chemical properties of oily sludge could be changed by adding the mixture of different functional chemical reagents as the modified materials, speeding up the natural semi-drying rate of oily sludge.

Because the natural semi-drying of oily sludge needs to control the economic cost, the water content in the same time after adding various treatment agents to oily sludge is different, mainly because different treatment agents have different purposes. The main role of desiccant:(1) react with water in oil sludge, consume water; (2) convert the emulsified water in the oil sludge into the oil and water separation state; (3) convert the adsorbed water and the combined water into free water in the oil mud; (4) Reaction exothermic, accelerating water evaporation; (5) The oxidation reaction reduces the organic matter in the oil and reduces its adsorption ability to water; (6) Other reagents can speed up the reaction time of inorganic gel materials or increase water absorption. Based on the above design, the mixture of the desiccant includes powder emulsifier, oxidant, inorganic gel material.

Thank you very much for your suggestion. I have added the selection and function of OSM materials to the experimental materials and added the corresponding references.

Reviewer #5: 

Finally, authors must clean up language and clarify to reader the different forms of water discussed in the manuscript. This should include a clear description of each term used and consistent use of this terminology throughout the manuscript. There are currently over ten different terms used throughout the manuscript. (currently in use: free water, gap water, surface water, combined water, internal free water, pore water, absorbed water, combined water, water in emulsion, interstitial water, free emulsion…)

Reply:

Thanks for your comments. Your suggestion is greatly appreciated. The water in oily sludge mainly exists in three forms: free water, bound water, and emulsified water. Free water refers to water that can flow freely, and bound water refers to the attraction of electrically charged molecules to the internal pore surfaces of oily sludge particles. Water on the external surface, emulsified water refers to oil-water emulsion, which is a water-in-oil, oil-in-water or multiple emulsified dispersion system.( This paragraph has been added to the end of the first paragraph)

This guidance had been very helpful to me, the full text has been revised, the main forms of water for free water, bound water, emulsified water, surface water to oil-bearing sludge particles surface bound water, internal Free Water is modified to free water between oily sludge particles, emulsion water to emulsified water, interstitial water to bound water between internal pores of oily sludge particles, and free emulsion to emulsified water. Here are some changes about different forms of water, more details please see the manuscript marked in red.

Line135, the “absorbed water” was changed into “bound water”;

Line 421, the “emulsion water” was changed into “emulsified water”;

Line500, the “free emulsion water” was changed into “emulsified water”;

Line503, the “interstitial water” was changed into “bound water between internal pores of oily sludge particles”;

Line509, the “interstitial water” was changed into “oil-bearing sludge particles surface-bound water”.

Reviewer #5: 

While the English of the manuscript is generally good there are still numerous typographical errors and areas that could use clarification. In addition to addressing these errors I encourage authors to complete a thorough read through of the paper to ensure their intended meanings have not been changed through the editing process. While not complete, below is a list of some specific suggestions and questions I noted as I read through the manuscript.

Reply:

Thank you for your very detailed suggestions. I have checked and revised them item by item. The details are as "Response to reviewers".

---

## [Decision Letter · Decision Letter 1]

2 Dec 2020

PONE-D-20-21667R1

Facilitating the natural semi-drying of oily sludge by changing the form of water

PLOS ONE

Dear Dr. Yucheng Liu, 

Thank you for submitting your manuscript to PLOS ONE. After careful consideration, we feel that it has merit but does not fully meet PLOS ONE’s publication criteria as it currently stands. Therefore, we invite you to submit a revised version of the manuscript that addresses the points raised during this next review process (see attachment file for most recent review comments).

We look forward to receiving your revised manuscript.

Kind regards,

David Rider

Academic Editor

PLOS ONE

Reviewer's Responses to Questions

**Comments to the Author**

1. If the authors have adequately addressed your comments raised in a previous round of review and you feel that this manuscript is now acceptable for publication, you may indicate that here to bypass the “Comments to the Author” section, enter your conflict of interest statement in the “Confidential to Editor” section, and submit your "Accept" recommendation.

Reviewer #1: (No Response)

2. Is the manuscript technically sound, and do the data support the conclusions?

Reviewer #1: Yes

3. Has the statistical analysis been performed appropriately and rigorously? 

Reviewer #1: Yes

4. Have the authors made all data underlying the findings in their manuscript fully available?

Reviewer #1: Yes

5. Is the manuscript presented in an intelligible fashion and written in standard English?

Reviewer #1: No

6. Review Comments to the Author

Reviewer #1: Please see attached for detailed comments and suggestions for the reviewed manuscript. See attached.

7. PLOS authors have the option to publish the peer review history of their article (what does this mean?). If published, this will include your full peer review and any attached files.

Reviewer #1: No

---

## [Author Response · Author response to Decision Letter 1]

19 Dec 2020

Thank you for your letter and for the reviewers’ comments concerning our manuscript entitled “. Those comments are all valuable and very helpful for revising and improving our paper. We have studied comments carefully and have made correction which we hope to meet with approval.We have revised the comments one by one, and also asked people outside the laboratory to help revise them, and finally carried out the native language polishing. Revised portion are marked in red or blue in the paper. We want our identity to be public for this peer review.

---

## [Editor Report · Decision Letter 2]

4 Jan 2021

Facilitating the natural semi-drying of oily sludge by changing the form of water

PONE-D-20-21667R2

Dear Yucheng Liu,

We’re pleased to inform you that your manuscript has been judged scientifically suitable for publication and will be formally accepted for publication once it meets all outstanding technical requirements.

Kind regards,

David Rider

Academic Editor

PLOS ONE

---

## [Editor Report · Acceptance letter]

6 Jan 2021

PONE-D-20-21667R2 

Facilitating the natural semi-drying of oily sludge by changing the form of water 

Dear Dr. Liu:

I'm pleased to inform you that your manuscript has been deemed suitable for publication in PLOS ONE. Congratulations! Your manuscript is now with our production department. 

Kind regards, 

on behalf of

Professor David Rider 

Academic Editor

PLOS ONE